# Improvement of the Antimicrobial Activity of Oregano Oil by Encapsulation in Chitosan—Alginate Nanoparticles

**DOI:** 10.3390/molecules26227017

**Published:** 2021-11-20

**Authors:** Krassimira Yoncheva, Niko Benbassat, Maya M. Zaharieva, Lyudmila Dimitrova, Alexander Kroumov, Ivanka Spassova, Daniela Kovacheva, Hristo M. Najdenski

**Affiliations:** 1Faculty of Pharmacy, Medical University of Sofia, 1000 Sofia, Bulgaria; krassi.yoncheva@gmail.com (K.Y.); nbenbassat@pharmfac.mu-sofia.bg (N.B.); 2The Stephan Angeloff Institute of Microbiology, Bulgarian Academy of Sciences, 1113 Sofia, Bulgaria; zaharieva26@yahoo.com (M.M.Z.); lus22@abv.bg (L.D.); adkrumov@gmail.com (A.K.); 3Institute of General and Inorganic Chemistry, Bulgarian Academy of Sciences, 1113 Sofia, Bulgaria; ispasova@svr.igic.bas.bg (I.S.); didka@svr.igic.bas.bg (D.K.)

**Keywords:** oregano oil, chitosan—alginate nanoparticles, antimicrobial activity, in vitro cytotoxicity, skin irritation test

## Abstract

Oregano oil (OrO) possesses well-pronounced antimicrobial properties but its application is limited due to low water solubility and possible instability. The aim of this study was to evaluate the possibility to incorporate OrO in an aqueous dispersion of chitosan—alginate nanoparticles and how this will affect its antimicrobial activity. The encapsulation of OrO was performed by emulsification and consequent electrostatic gelation of both polysaccharides. OrO-loaded nanoparticles (OrO-NP) have small size (320 nm) and negative charge (−25 mV). The data from FTIR spectroscopy and XRD analyses reveal successful encapsulation of the oil into the nanoparticles. The results of thermogravimetry suggest improved thermal stability of the encapsulated oil. The minimal inhibitory concentrations of OrO-NP determined on a panel of Gram-positive and Gram-negative pathogens (ISO 20776-1:2006) are 4–32-fold lower than those of OrO. OrO-NP inhibit the respiratory activity of the bacteria (MTT assay) to a lower extent than OrO; however, the minimal bactericidal concentrations still remain significantly lower. OrO-NP exhibit significantly lower in vitro cytotoxicity than pure OrO on the HaCaT cell line as determined by ISO 10993-5:2009. The irritation test (ISO 10993-10) shows no signs of irritation or edema on the application site. In conclusion, the nanodelivery system of oregano oil possesses strong antimicrobial activity and is promising for development of food additives.

## 1. Introduction

The antimicrobial resistance to clinically approved antibiotics and chemotherapeutics rapidly growing all over the world warrants the search for alternative sources of antimicrobial compounds, both for the prevention and treatment of infectious diseases in human and veterinary medicine and for the preservation of food products in the food industry. In this aspect, the use of modern nanotechnological approaches is essential for the development of new products with known antimicrobial ingredients aiming to increase their effectiveness. Oregano (*Origanum vulgare* L.) is a herb prominent in the Mediterranean and particularly in the Bulgarian diet. It has been shown to possess numerous bioactive properties such as antimicrobial, anti-inflammatory, antioxidant, and analgesic properties [1,2]. According to ethnopharmacological data, oregano oil (OrO) has been known since ancient times as the strongest natural antibiotic and has been used successfully in traditional medicine to relieve diseases of the gastrointestinal tract (e.g., diarrhea, indigestion, stomachache), diabetes, respiratory (e.g., asthma, bronchitis, cough), infectious, inflammatory, and menstrual disorders [2,3,4]. In addition, it is widely used as a spice in cooking and is part of many food products on the food market [1]. OrO from *Origanum vulgare* is currently authorized as a feed additive according to the entry in the European Union Register of Feed Additives pursuant to Regulation (EC) No. 1831/2003 [5].

Recent preclinical pharmacological studies on the therapeutic properties of OrO and both of the main molecules in its composition, the terpenoid thymol and its phenol isomer carvacrol, have demonstrated a wide spectrum of pharmacological activities such as antibacterial [6,7,8,9,10,11,12,13,14,15,16,17,18,19,20,21,22,23,24], antifungal [22,23,25,26], antiviral [27,28,29,30], antioxidant, etc. According to the published data, OrO, carvacrol, and thymol exert a bacteriostatic or bactericidal effect in a concentration-dependent manner against food-borne pathogens, human or animal clinical isolates, including a broad spectrum of Gram-positive (*Staphylococcus* spp., *Streptococcus* spp., *Enterococcus* spp., *Acinetobacter* spp., *Bacillus* spp., *Listeria monocytogenes*, etc.) and Gram-negative (*Escherichia coli*, *Pseudomonas aeruginosa*, *Klebsiella* spp., *Salmonella* spp., *Propionibacterium acnes*, etc.) species.

Undoubtedly, the most widespread current application of OrO is in the food industry because of its bactericidal effect against a wide range of food-borne pathogens. Despite all different techniques used in food industry for ensuring food safety and quality, food-borne diseases are still a serious public health problem [10]. For the last two decades, four food-borne bacterial species represent a major problem in Europe and are in the focus of government agencies and the food industry: *Salmonella* spp., *E. coli*, *Campylobacter* spp., and *Yersinia* spp. [31,32]. An important challenge for the food industry is also the increased use of ready-to-eat, ready-to-cook, and heat-and-eat foods in industrialized countries. The new trend and regulations for the production of all-natural food without artificial preservatives which may have toxic properties or cause allergies require new approaches for food preservation [12,33]. Many studies report on the activity of OrO against food-borne pathogens which together with its meat seasoning properties makes it a suitable candidate for a food preservative agent [1,10,12,20,22]. Boskovic et al. [20] demonstrated that OrO inhibited the growth of *S. typhimurium* and *B. cereus* at the concentration of 160 µg/mL, whereas *S. enteritidis*, *E. coli*, and MRSA were susceptible to 320 µg/mL. Pesavento et al. [12] showed that a 0.5% OrO concentrate exerted a bacteriostatic effect on *L. monocytogenes* in raw minced meat up to the 11th day and a bactericidal effect after 14 days of incubation. Greater concentrations of 1–2% OrO killed the bacteria on the second day of the experiment. In the same study, the MICs for *L. monocytogenes*, *S. aureus*, *S. enteriditis*, and *Campylobacter jejuni* were in the range of 0.006–0.05% as determined by the broth microdilution method. Cattelan et al. tested in their study four food-borne pathogens—*B. cereus*, *B. subtilis*, *E. coli*, and *S. typhimurium* and reported that all these species were susceptible to OrO, but the highest concentration (5%) was most efficient against *B. cereus* [10]. Cattelan et al. [34] studied the effects of OrO on the survival and growth of *E. coli* in salad dressings and demonstrated that the use of OrO can be considered promising as far as it allows reduction in the bacterial count and the levels of salt to be incorporated in food. OrO was also tested as an additive to different meat products such as chorizo [35] and tsamarella [36] in concentrations of 0.1% and 1%, respectively. According to the published data, OrO did not affect the growth of the starter cultures in both meat products and decreased the bacterial pathogen count (*Enterobacteriaceae*) in tsamarella. OrO also significantly reduced the heat treatment time required for the inactivation of *L. monocytogenes* in products containing sous-vide salmon [37]. Based on these reports, OrO can be considered a healthy ingredient for food products where applicable but its application is often limited due to flavor considerations since effective antimicrobial doses may exceed organoleptically acceptable levels [7,12,36]. In general, it was found that concentrations between 0.1% and 1% are suitable regarding the taste for meat preservation [12,35,36]. The use of OrO as a food additive has already been recognized in the European Union in the concentrations of 2.0 mg/kg bw/day for women and 2.3 mg/kg bw/day for men, respectively [1,38]. For the purposes of the food industry, OrO is often not added to the product itself, but is incorporated into biocomposite packaging films or other types of edible coatings. Edible coatings based on polysaccharides are environmentally friendly alternatives and can reduce the use of chemical preservatives [39,40,41,42,43,44]. The application of such packages is investigated for numerous types of raw or processed foods such as pork, ham, chicken wingettes, fish fillets, seafood, vegetables, fruits, juices, etc. [39,40,45,46,47,48,49,50,51,52]. The incorporation of 2% OrO in a chitosan-based biocomposite film achieved a growth inhibition rate of 99% against *E. coli* and *L. monocytogenes* [41]. Edible Na alginate films with OrO significantly reduced the count or killed *L. monocytogenes* in packages with ham slices.

In general, application of the oregano oil is limited to formulation in lipid vehicles, mainly for external treatment. On the other hand, oregano oil is volatile and susceptible to degradation, which additionally hinders its pharmaceutical application. The aim of our study was to develop a nanosized system that would either enable formulation of oregano oil in an aqueous dispersion or potentiate its antimicrobial effect. Chitosan and sodium alginate were selected as vehicles of the nanosystem because of their biocompatibility and the antibacterial properties of chitosan itself. Further, part of the studies was focused on the evaluation of in vitro cytotoxicity of the developed nanodelivery system in normal human keratinocytes and in vivo skin irritation test on rabbits.

## 2. Results

### 2.1. Phytochemical Characterization of the Oregano Oil

Oregano oil was obtained by distillation using a Clevenger apparatus with a yield of 1.5%. Characterization of the chemical constituents of the essential oil was performed by thin-layer chromatography and gas chromatography–mass spectrometry (GC–MS). After calculating the area under the curve using the Genesis algorithm in the NIST program, the relative concentrations of the main compounds in the oil were defined (Table 1).

### 2.2. Physicochemical Characterization of Oregano Oil-Loaded Chitosan—Alginate Nanoparticles

The XRD patterns of the oregano oil, the empty and the oil-loaded nanoparticles are presented in Figure 1. It can be seen that the empty nanoparticles showed very low crystallinity with only one broad diffraction peak centered at around 23° 2θ. The oregano oil showed two diffraction peaks, the stronger one at 18° 2θ and the weaker and broader one at 42° 2θ. The OrO-loaded nanoparticles (OrO-NP) presented two peaks at about 20° and 42° 2θ. The shift of the strongest peak toward higher 2θ is an indication of the shortening of the interatomic distances in the OrO-loaded nanoparticles, pointing to the formation of a denser amorphous network compared to the pure oil. This also implies that incorporation of oil into the nanoparticles might result in a change in the chitosan–alginate packing structure.

### 2.3. FTIR Spectra of Pure Oregano Oil, Empty and OrO-Loaded Chitosan—Alginate Nanoparticles

The FTIR spectra of the pure oregano oil, empty and OrO-loaded chitosan—alginate nanoparticles are presented in Figure 2. The oregano oil showed a broad band at ~3380 cm^−1^ (Figure 2) which was assigned to the O–H stretching vibration [4,53,54]. The three bands between 2870 and 2960 cm^−1^ were attributed to the C–H stretching vibration of aliphatic CH_2_– groups. Between 1600 cm^−1^ and 1300 cm^−1^, several bands attributed to the C–H bending of the aliphatic CH_2_ groups and C–O–H bending are visible [55]. The other characteristic peaks (confirming the presence of carvacrol and thymol) appeared in the fingerprint region (900–1200 cm^−1^) in agreement with those found by Valderrama et al. in 2017 [56]. The band around 812 cm^−1^ could be attributed to out-of-plane C–H wagging vibrations, the most significant signal used in distinguishing between different types of aromatic ring substitution [53]. The spectrum of the empty chitosan—alginate nanoparticles is presented in Figure 2b. As the quantity of the chitosan is much lower than that of the alginate, the FTIR spectrum of the nanoparticles is much closer to those of pure alginate than to chitosan [57]. One can observe a broad band at 3415 cm^−1^ corresponding to the stretching vibrations of the O–H and N–H bonds originating from amine and hydroxyl groups. Peaks at 2927 cm^−1^ and 2850 cm^−1^ are due to C–H stretching. The peak at 1739 cm^−1^ corresponds to C=O stretching modes of COOH [57]. Characteristic absorption bands of chitosan are usually seen between 1649 and 1652 cm^−1^ and 1558–1598 cm^−1^, corresponding to C–O stretching (amide I) and N–H bending (amide II), respectively [58,59,60]. When reacting with alginate, these bands shift and overlap each other, creating a strong peak at 1611 cm^−1^ [61]. The bands at 1616 cm^−1^ and 1421 cm^−1^ were assigned to asymmetric and symmetric stretching vibration of carboxylate groups. Due to the presence of polysaccharide structure, bands around 1300 cm^−1^ (C–O stretching), 1128 cm^−1^ (C–C stretching), 1084 cm^−1^ (C–O stretching), 1034 cm^−1^ (C–O–C stretching), and 946 cm^−1^ (C–O stretching) are visible, too [60]. In the spectrum of oregano oil-loaded nanoparticles (Figure 2c), the characteristic peaks between 2870 and 2960 cm^−1^ appear at the same wave number indicating the successful encapsulation of the oil into the polymer matrix [62]. However, an interaction between the oregano oil and the chitosan—alginate nanoparticles is not excluded because of the changed intensity and the appearance of new peaks. Similar interaction is observed for microencapsulated oregano oil using acacia, starch, and maltodextrin [63].

### 2.4. Thermogravimetric Analysis of Pure Oregano Oil, Empty and OrO-Loaded Chitosan—Alginate Nanoparticles

The oregano oil, empty and loaded chitosan—alginate nanoparticles were examined by thermogravimetric analysis. The temperatures corresponding to the maximum weight loss step could be seen from the first derivative of the TGA curve (DTG) on the temperature. The results in the temperature range of 50–600 °C are presented in Figure 3. The pure oregano oil had one sharp step of weight loss, while the empty and the oregano oil-loaded nanoparticles showed two steps of weight loss. The pure oregano oil decomposed completely at 220 °C. The empty chitosan—alginate nanoparticles decomposed in two steps, at 65 °C and at 245 °C. For the oil-loaded nanoparticles, the first slight peak was centered at about 250 °C and the main degradation peak could be seen at a higher temperature, 395 °C. This observation was in accordance with previously reported results [62,64]. The authors pointed out that the encapsulated carvacrol in chitosan particles decomposed at a higher temperature than free carvacrol. Hence, the encapsulation of the oregano oil into the nanoparticles led to an improved thermal stability. This effect could be due to the possible interaction between the essential oil and the nanoparticles.

### 2.5. Minimal Inhibitory and Bactericidal Concentrations of Pure OrO and OrO-Loaded Chitosan—Alginate Nanoparticles

The tested OrO-NP exhibited strong antimicrobial activity against all the tested pathogenic microbial strains (Table 2). The MICs for the Gram-positive and Gram-negative bacteria varied between 0.06% and 1% *v*/*v*, wherein the *S. aureus*, *E. faecalis*, *E. coli*, and *Y. enterocolitica* strains showed equal susceptibility to OrO. *S. pyogenes* and *P. aeruginosa* were less sensitive. The growth of the fungal strain *Candida albicans* was inhibited after exposure to 0.125% *v*/*v* of OrO, which is in the same concentration range determined for the bacterial strains. The MBC values were the same as the MIC values for *S. pyogenes* and *Y. enterocolitica*. The other MBC values were twofold (MRSA, *C. albicans*), fourfold (*S. aureus*), and eightfold (*E. faecalis*) greater than the respective MIC values. The OrO-NP formulation was characterized by a significantly greater activity than pure oregano oil regarding bacterial growth inhibition expressed in up to tenfold lower MIC values (Table 2, Figure 4).

### 2.6. Dehydrogenase Activity of Pathogenic Bacteria Treated with Pure OrO and OrO-Loaded Chitosan—Alginate Nanoparticles

OrO diminished the dehydrogenase activity (DEHA) of all the treated bacterial species and the *C. albicans* strain by more than 97%. Only three strains—*E. faecalis*, *S. pyogenes*, and *Y. enterocolitica* retained 16.94%, 11.60%, and 7.05% of their metabolic activity, respectively. A graphical comparison between the DEHA activities of OrO and OrO-NP is presented in Figure 4.

The DEHA activity of the microbial strains is presented in Figure 5 as the relationship between the applied concentrations and the respective inhibitory effects. The coefficients of inhibition varied between 0.0002% (*P. aeruginosa*) and 0.1997% (*S. pyogenes*). An inversely proportional relationship was observed between the estimated MICs and the corresponding DEHA activity. OrO-NP inhibited the DEHA activity of all the strains at concentrations higher than 0.01% excluding *E. faecalis* and *S. pyogenes* which turned out to be less susceptible even to the highest concentrations applied.

### 2.7. In Vitro Cytotoxicity of OrO and OrO-Loaded Nanoparticles

The median inhibitory (IC_50_) and maximum nontoxic concentrations (MNT) for OrO, OrO-NP, and chitosan–alginate are presented in Figure 6. Concerning the IC_50_ values, OrO-NP are significantly less cytotoxic than OrO. Chitosan–alginate is not toxic to the HaCaT cells in concentrations of the nanodelivery system achieving antimicrobial activity. The MNT concentrations of OrO and OrO-NP are equal.

### 2.8. Effects of Pure OrO and OrO-Loaded Chitosan—Alginate Nanoparticles on Skin Irritation in Rabbits

OrO and OrO-NP were applied for 4 h onto rabbits and their dermal safety was assessed at 24, 48, and 72 h after the exposition period (Figure 7). The Primary irritation score (PIS) and the primary irritation index (PII) for OrO and OrO-NP were equal to zero. In comparison, PIS and PII of the positive control (10% SDS) were equal to 3. As long as a single exposure to OrO or OrO-NP did not lead to skin irritation, the cumulative irritation index (CII) was not calculated.

## 3. Discussion

The GC–MS analysis showed that the distilled oregano oil contained previously reported compounds like carvacrol, thymol, γ-terpinene, and α-pinene [15,65]. The interesting fact is that instead of p-cymene, which is a frequently reported compound, in our oil sample, o-cymene and m-cymene were detected. In addition, the percentage of terpinolene in the oil was higher compared with the percentages reported in other studies [66]. Furthermore, the other compounds present in the oil were bergamol and aromadendrene. Taking into account all the mentioned compounds, especially the high concentration of carvacrol in the oil, a strong antibacterial activity could be expected. Aiming to enable its application in therapy and food processing, the oregano oil was encapsulated in chitosan—alginate nanoparticles. The loading of the oil into chitosan—alginate nanoparticles was carried out by emulsification of the oil into an aqueous solution of sodium alginate and consequent electrostatic gelation with chitosan. The method resulted in 50% encapsulation efficiency. Both the X-ray diffraction and IR spectroscopy proved successful encapsulation of the oil into the nanoparticles. On the other hand, thermogravimetric analysis revealed that the encapsulation of the oregano oil into the nanoparticles resulted in higher thermal stability, which indicated that the nanoparticles could stabilize the oil. The oil-loaded nanoparticles were characterized with a mean diameter of approximately 320 nm, polydispersity index of 0.631, and a negative zeta-potential (−25 mV). Thus, the small size and the magnitude of the negative zeta-potential provide good colloidal stability of the dispersion. This is in agreement with reports that values of zeta-potential ± 30 mV indicate high stability due to electrostatic repulsion [67].

The comparison between the antimicrobial potential of the pure and encapsulated oregano oil revealed that the nanoformulation inhibited bacterial growth more efficiently at lower concentrations than the pure oil. The MICs of OrO-NP were from four- up to 32-fold lower than the MICs of the OrO confirming higher effectiveness of the nanoparticles (Table 2). Concerning the MBC values, the OrO-NP solution exhibited a stronger bactericidal efficiency on both *Staphylococcus* strains, whereas for the other bacterial test strains, both formulations were characterized with an approximately equal antibacterial potential. Our results for the antibacterial activity of the pure OrO are in line with numerous experimental data of other authors. The effective concentrations of OrO estimated in several published studies varied between 1.25–1600 µg/mL [7,10,16,30,68,69]. Presented in percentage, the minimal inhibitory concentration (MIC) of OrO ranged depending on the strain from 0.05% to 6.23% [8,18] in broth model systems (test microorganisms *S. aureus*, *E. faecalis*, *E. coli*, *K. pneumonia*, and *P. aeruginosa*) and from 1% to 5% (a large set of food-borne pathogens) when the disk diffusion method was preferred as the evaluation model [10]. A recent study by Sobczyk et al. [24] reported high antibacterial activity of oregano oil and oregano leaves on chitosan–alginate-based dressings for the treatment of wounds. The active concentrations of oregano oil on *S. aureus* and *E. coli* were 0.25% and 0.5% for both strains whereas oregano leaves exerted an equivalent antimicrobial potential at the concentrations of 10% and 20%. As published by Sim et al. [19], a total of 80 bacterial and fungal isolates (including *S. pseudintermedius*, β-hemolytic *Streptococcus* spp., *P. aeruginosa*, and *P. mirabilis*) from cases of canine otitis externa were investigated for susceptibility to OrO and carvacrol, and the determined MICs of both ranged between 0.05% and 0.125%, which corresponds to the effective concentrations in our study.

The data analysis of the inhibition of respiratory activity (Table 2) revealed the excellent descriptive characteristics of the selected Lambert–Pearson mathematical model for the evaluation of biochemical processes at the population level. It can be successfully adapted and used for a wide range of substances in antimicrobial susceptibility testing of bacterial pathogens. The respiratory activity of the tested bacteria was higher after exposure to OrO-NP (ranging between 0.34% and 91.84% at the MICs) than to OrO (between 2.09% and 16.94%), except for the *Staphylococcus* strains and *P. aeruginosa* whose metabolism was significantly inhibited by OrO-NP. This can explain why the MBC values for OrO-NP were the same as for OrO. From the data obtained it can be assumed that the lower effective concentrations of OrO-NP have a bacteriostatic effect, while to achieve a bactericidal effect, it is necessary to apply higher concentrations of oil regardless of the application form. It is noteworthy to discuss that *Yersinia enterocolitica* was one of the most sensitive bacterial species tested. According to the reports of EFSA on zoonoses, yersiniosis was the fourth mostly reported zoonosis in humans in 2019, with a stable trend in 2015–2019 [70]. *Yersinia enterocolitica* is found predominantly in meat and meat products. As long as oregano is a commonly used spice for meat due to its favorable taste properties, the use of oregano oil as an antimicrobial preservative in meat products or incorporated into biocomposite packaging films would be particularly appropriate for food preservation purposes. Our results for the active concentrations of oregano oil confirm the results of other authors as the minimum inhibitory concentrations determined in our study fall into the same range of effectiveness (0.05% to 6.23%) for *S. aureus*, *E. faecalis*, *E. coli*, and *P. aeruginosa* in broth model systems [8,18]. The effective concentrations determined in our study are all below 0.5%. This concentration was shown by Pesavento et al. [12] to be suitable for food preservation as far as it does not substantially alter the flavor of the meat and still exerts an antibacterial effect against the quantity of pathogens usually found in raw meat (10 CFU/g or lower).

Oregano oil is widely used in oral and topical pharmaceutical formulations. In human and veterinary medicine, OrO could be applied for the treatment of numerous disorders caused by bacterial and fungal pathogens such as food-borne infections, drug-resistant *E. coli*-associated infections, canine otitis externa, as an antimicrobial in the poultry industry, etc. [5,8,9,17,19,71,72]. In addition, it was successfully implemented in the food industry in package nanolayers to replace preservatives [1,20]. According to the scientific opinion of EFSA about the safety of *Origanum vulgare* ssp. *hirtum* on different animal species, the safe oral doses vary depending on the species between 22 and 150 mg/kg, and no concerns for consumer safety were identified for oral use up to the maximum safe concentrations in feed [5]. However, a possibility of skin irritation after application of products containing oregano oil was indicated in the report. Therefore, we assessed the contact hazard potential of the formulations with oregano oil investigated in our study following 1) the in vitro cytotoxicity test for medical devices (ISO 10993-5 (Annex V)) applied on the HaCaT cell line derived from normal human keratinocytes and 2) the skin irritation test on albino rabbits (ISO 10993-10). OrO-NP were significantly less cytotoxic (up to fourfold) than OrO when the median inhibitory concentrations were compared. Chitosan–alginate was not cytotoxic in the concentrations achieving the antimicrobial effect. Both formulations showed equal maximum nontoxic concentration in vitro. Nevertheless, neither formulation caused erythema or edema on the skin of the experimental animals. Thus, regarding the skin irritation potential of both formulations, we did not observe dermal irritation even at concentrations tenfold higher than the MICs.

## 4. Materials and Methods

### 4.1. Distillation and Chemical Composition of Oregano Oil

*Origanum vulgare* was cultivated at the area of Panagyurishte, a mountainous region northwest of Pazardzhik, Bulgaria. The essential oil was obtained by distillation using a Clevenger apparatus. A quantity of 30 g of dried *Origanum vulgare* leaves and stems was cut and crushed into pieces less than 1 × 1 cm that were subsequently immersed in approximately 500 mL of distilled water in a Clevenger apparatus. Distillation was run for about 3 h at medium-low heat until exhaustion. After completion of distillation, the oil was collected and stored in 10 mL dark glass bottles. The essential oil was left open under a filter cap for 24 h in order to evaporate any remaining solvent and after closing was stored in the fridge (5–7 °C).

Gas chromatography–mass spectroscopy (GC–MS) analysis was conducted according to Ph. Eur. 9th Edition [73]. The GC–MS analysis of diluted (1:1000) oregano oil was performed on an Exactive Orbitrap GC–MS system (ThermoFisher Scientific, Waltham, MA, USA) operating at 70 eV, ion source temperature of 230 °C, interface temperature of 280 °C, with split injection (1 μL, ratio 20:1) at the injector temperature of 270 °C. A fused silica capillary column, 5% phenyl / 95% methyl polysiloxane (TG-5SILMS 30 m × 0.25 mm × 0.25 μm, ThermoFisher Scientific, Waltham, MA, USA) was used. The carrier gas was helium 5.0 at the flow rate of 1.0 mL/min. Data acquisition was performed with Xcalibur 4 (40–600 u).

For the GC–MS analysis, raw data were imported and opened in the NIST Mass Spectral Search Program (Version 2.0 g) and the mass spectra were compared with those in the NIST/EPA/NIH Mass Spectral Database library (NIST 11). Finally, the Genesis algorithm included in the NIST Mass Spectral Search Program was used in order to detect the main peaks in the GC–MS chromatogram and quantify the percentage of the main chemical constituents.

### 4.2. Encapsulation of Oregano Oil

Sodium alginate (very low viscosity) was supplied by Alfa Aesar GmBH & CoKG (Karlsruhe, Germany). Chitosan (Mv 110 000–150 000) was provided by Sigma Aldrich. Encapsulation of oregano oil into chitosan—alginate nanoparticles was performed by emulsification and consequent electrostatic gelation of both biopolymers by adaptation of a reported procedure (Lertsutthiwong et al., 2009). First, oregano oil was dissolved in methylene chloride and emulsified in an aqueous solution of sodium alginate (3 mg/mL) containing Tween 80. Emulsification was performed under sonication for 2 min at 20 kHz (Bandelin Sonopuls, Berlin, Germany). Then, a calcium chloride solution (3.35 mg/mL) was added to the resultant o/w emulsion under sonication for 1 min. The emulsion was stirred for 30 min (300 rpm), and after that, a chitosan solution (0.75 mg/mL) was slowly dropped. The dispersion was stirred for 24 h and centrifuged (15 min, 20 000 rpm) and the pellets were rinsed with purified water.

Determination of oregano oil loading was made by means of UV–VIS spectrophotometry [63]. The standard curve of the oil was prepared by diluting its stock solution with methanol (5.31–63.8 µg/mL, r > 0.998) and spectrophotometric measurement of absorbance at λ = 275 nm (ThermoFisher Scientific, Waltham, MA, USA).

### 4.3. Characterization of Nanoparticles

Nanoparticle diameter, polydispersity, and zeta-potential were determined by photon correlation spectroscopy and electrophoretic laser Doppler velocimetry using a Zetasizer Nano Series (Malvern Instruments, Worcestershire, UK). The aqueous dispersions of the nanoparticles were measured at 25 °C with the scattering angle of 90°.

Thermogravimetric analyses of the oregano oil, empty and oregano oil-loaded nanoparticles were evaluated in dynamic conditions using LABSYSEvo, SETARAM (Caluire, France) in an argon atmosphere, with heating rates of 10 °C min^−1^, temperature interval, 30–650 °C.

IR spectra were recorded with a Nicolet iS5 FTIR spectrometer accumulating 64 scans at the spectral resolution of 2 cm^−1^.

Powder X-ray diffraction patterns of the oregano oil, empty and oregano oil-loaded nanoparticles were collected in the 2θ range of 5.3° to 80° with a step of 0.02° 2θ on a Bruker D8 Advance diffractometer with CuKα radiation equipped with a LynxEye detector. The diffraction pattern of doxorubicin was indexed using the Topas 4.2 program.

### 4.4. Bacterial and Fungal Strains and Culture Conditions

The bacterial strains used for determination of the antimicrobial activity of OrO and OrO-NP were *Staphylococcus aureus* (ATCC^®^ 29213^TM^, American Type Cell Culture Collection, Manassas, VA, USA), *Staphylococcus aureus*—MRSA (NBIMCC 8327—resistant to methicillin and oxacillin, National Bulgarian Collection for Industrial Microorganisms and Cell Cultures), *Streptococcus pyogenes* (SAIMC 10535, Collection of the Stephan Angeloff Institute of Microbiology, Sofia, Bulgaria), *Enterococcus faecalis* (ATCC^®^ 29212^TM^, Manassas, VA, USA), *Escherichia coli* (ATCC^®^ 35218^TM^, Manassas, VA, USA)*, Pseudomonas aeruginosa* (ATCC^®^ 27853^TM^, Manassas, VA, USA) and *Yersinia enterocolitica* (IP864 O:3, Collection of Institut Pasteur, Paris, France). The antimycotic activity was tested on *Candida albicans* (CBS 562, Utrecht, The Netherlands)*. S. aureus*, MRSA, *E. coli*, and *E. faecalis* were grown in Mueller Hinton broth (MHB, #M0405B, Thermo Scientific-Oxoid, Hampshire, UK) and agar (MHA, #CM0337, Thermo Scientific-Oxoid, Hampshire, UK); *S. pyogenes*, *P. aeruginosa*, *Y. enterocolitica*, and *C. albicans* were grown in brain heart infusion broth (#M210, Himedia, Mumbai, India) and agar (BHIA). MHA and BHIA were prepared by adding the respective concentration of agar (#RM10848, Himedia, Mumbai, India).

### 4.5. Determination of MIC and MBC

The MICs of OrO and OrO-NP were determined according to ISO 20776-1:2006 [74] based on the broth microdilution method (BMD). Twofold serial dilutions of both substances ranging from 0.0005% up to 1% volume fraction were prepared for each microbial strain in 96-well plates (50 µL/well) in threefold repetitions. An equivalent volume of the bacterial suspension with the density of 1 × 10^5^ CFU/mL (prepared using the McFarland standard) was added to each well. The plates were incubated at 37 °C for 24 h. The lowest drug concentration, which inhibited the visible bacterial growth was accepted as MIC. Gentamicin (0.008–4 mg/L) and penicillin (0.008–4 mg/L) were used as the reference antibiotics (positive control). The recommendations of EUCAST (European Committee on Antimicrobial Susceptibility Testing) were followed for the analysis of the results [75]. PBS served as the negative control, whereas MHB and MHB with OrO or OrO-NP—as the blank solutions. The MBCs were determined after seeding of the samples treated from 1/2× MIC up to the highest concentration on Petri dishes with MHA and the diameter of 9 cm. These samples were incubated at 37 °C for 24 h. MBC was defined as the lowest drug concentration reducing colony growth of the initial bacterial inoculum by ≥99.9%.

### 4.6. Determination of the Dehydrogenase Activity

The dehydrogenase (respiratory, metabolic) activity of the bacterial strains after treatment with OrO and OrO-NP was measured using the MTT dye, the reduction of which to formazan crystals was catalyzed by the membrane-located bacterial enzyme NADH:ubiquinone reductase (H^+^-translocation). The protocol of Wang et al. [76] was applied after minor modifications. Briefly, the inoculums of the test strains and the serial dilutions of the test substances were prepared as described for the BMD assay and the treated samples were incubated for 24 h at 37 °C. Thereafter, 10 µL of the MTT solution (5 mg/mL) were added to each well and the plates were left at 37 °C for 120 min. The resulting non-soluble violet formazan crystals were dissolved with an equivalent volume of 2-propranol containing 5% formic acid. The absorbance was measured at 550 nm (Absorbance Microplate Reader Lx800, Bio-Tek Instruments Inc., Winooski, VT, USA) against a blank solution containing the respective volumes of MHB and MTT. The self-absorbance of OrO and OrO-NP was also subtracted from the measured values.

### 4.7. Cell Viability Assay

The MTT test was performed according to Annex C to ISO 10993-5 [77,78] in order to determine the median inhibitory (IC_50_) and MTC concentrations of the solvents and PAC. The nontumorigenic HaCaT cell line (immortalized human keratinocytes, CLS Cell Lines Service GmbH, Eppelheim, Germany) was used as an in vitro model for skin cytotoxicity. The cells were maintained under sterile conditions in a CO_2_ incubator (Panasonic MCO-18AC, Kadoma, Osaka, Japan) supplying 5% CO_2_ at 37 °C and humidified atmosphere. For the cultivation of the cell line, the culture medium DMEM-HG (#DMEM-HPA, Capricorn^®^, Munich, Germany) was used, supplemented with 4.5 g/L glucose, 10% heat-inactivated fetal bovine serum (#FBS-HI-12A, Capricorn^®^, Munich, Germany), 4 mM L-glutamine (#G7513, Sigma-Aldrich, Steinheim, Germany), and 10^5^ U/L penicillin G sodium and 100 mg/L streptomycin sulphate (Pen/Strep, #PS-B, Capricorn^®^, Munich, Germany). For the cell viability assay, a volume of 100 µL cell suspension with the density of 1 × 10^5^ cells/mL was seeded per well in 96-well plates with the flat bottom. The plates were incubated for 24 h until the cells entered the log phase of their growth. Thereafter, they were treated with OrO, OrO-NP, or chitosan–alginate in the concentrations ranging from 0.0002% to 0.1% *v*/*v* for OrO or from 0.0002% to 0.02% *v*/*v* for OrO-NP and chitosan–alginate. The result was read at the 72nd hour of the exposure to the tested formulations after dissolving the MTT product formazan in 100 μL/well organic solvent (% formic acid in 2-propanol). The same solvent was used as the blank. Untreated cells served as the negative control. The absorption was measured at λ = 540 nm/_ref_690 nm on a microplate reader ELx800 (BioTek Instruments, Inc., Winooski, VT, USA).

### 4.8. Mathematical Modeling Calculation of Respiratory (Metabolic) Activity

The quantitative evaluation of the inhibition effects of OrO and OrO-NP on the test microbial strains was performed using the Lambert–Pearson (LP) model considering the inhibitory effect of the applied concentration and fitting the experimental data on the base of the weighted least squares statistical method [79,80]. The program was coded in the MAPLE symbolic mathematics software. The model can be represented as follows:(1)$$Fa=\exp\left[-{\left(\frac{Dose}{P1}\right)}^{P2}\right]$$
where *Fa* stands for the normalized maximum enzyme activity, %; *Dose* stands for the concentration of the drug, µmol/L; *P*1 is the inhibitory constant, which may be interpreted as IC_50_ in medical studies, µmol/L; and *P*2 stands for the slope.

### 4.9. Calculation of the Median Inhibitory Concentrations and RSA Analysis

The calculation of the median inhibitory concentrations (IC_50_) was performed as published before [81]. Briefly, a nonlinear regression procedure based on the weighted least squares statistical criterion as an objective function of the search was coded in the MAPLE^®^ software of symbolic mathematics. The sum of weighted squares was minimized and the estimation of the best-fitting parameter values was found by using a numerical optimization algorithm. The median inhibitory concentration model was applied to calculate *IC*_50_ and *m*, as presented in Equation (1):(2)FaFu=(DoseDm)m
where *F_a_* is the affected fraction; *F_u_*—the unaffected fraction (1 – *F_a_*) = *F_u_*; *Dose*—the applied concentration; *D_m_*—the median inhibitory concentration (*D_m_ = IC_50_*); and *m*—the hillslope (for *m =* 1, the curve is hyperbolic; for *m* > 1—sigmoidal; for *m* < 1—negative (flat) sigmoidal). The response surface analysis (RSA) methodology was applied to reveal the predictive power of the model as a function of the parameters *IC*_50_ and *m*. The range of the value changes in the RSA 3D plot was determined based on the standard deviation of the *IC*_50_ and *m* values obtained during the statistical evaluation of the experimental data with the GraphPad Prism software.

### 4.10. Skin Irritation Test

The potential of the oregano oil used to produce dermal irritation was assessed according to the standard protocol in ISO 10993-10 [82]. Permit to work with experimental rabbits No. 232 for animal house No. 1113-0005 was issued by the Ministry of Agriculture, Food, and Forestry, Bulgarian Food Safety Agency, and valid until 11.04.2024. Briefly, three healthy young albino rabbits with intact skin were used as the test animals. The animals were acclimatized and cared for as specified in ISO 10993-2 [83] and Ordinance No. 20 (State Gazette of Bulgaria, No. 87, 09.11.2012). The fur on the back of the animals was clipped (10 × 15 cm) 4 h before the test. An amount of 0.5 mL of the test or control material was applied directly to the skin and covered with an absorbent gauze patch. Sunflower oil and 10% SDS solution were used as the negative and positive controls, respectively. The application site was wrapped with a semi-occlusive bandage for 4 h. Thereafter, the positions of the sites were marked with permanent ink. The residual test material was removed with lukewarm water and the skin was carefully dried. The reaction of each application site was recorded at the 1 (±0.1) h, 24 (±2) h, 48 (±2) h, and 72 (±2) h after removing the nonocclusive dressings. The skin reaction was described and scored for erythema and/or edema according to the scoring system given in ISO 10993-10. The primary irritation index (PII) was calculated based on the primary irritation score (PIS) for each sample and the results were read based on the scoring system for skin reaction.

## 5. Conclusions

Chitosan—alginate nanoparticles are considered an appropriate nanosized system for oregano oil. The studies revealed improved antibacterial activity of encapsulated oregano oil. Interestingly, the encapsulated oregano oil had a significantly higher antimetabolic activity than the pure oil in six of the eight tested strains. Based on these results, the oregano oil-loaded nanosystem is promising in terms of further development as a food additive with antimicrobial activity. In particular, the proposed nanodelivery system should be further screened for the antibacterial effect in vacuum-packed foods like meat, cheese, etc., accounting for their compatibility with the oregano oil taste and other organoleptic characteristics.

On the other hand, in vitro cytotoxicity on human keratinocytes and in vivo skin irritation test on rabbits demonstrated the safety profile of the nanoparticle formulation. Thus, the developed oregano oil-loaded chitosan—alginate nanoparticles could be considered an appropriate topical delivery system for treatment of microbial infections of skin and other soft tissues.

## Figures and Tables

**Figure 1 molecules-26-07017-f001:**
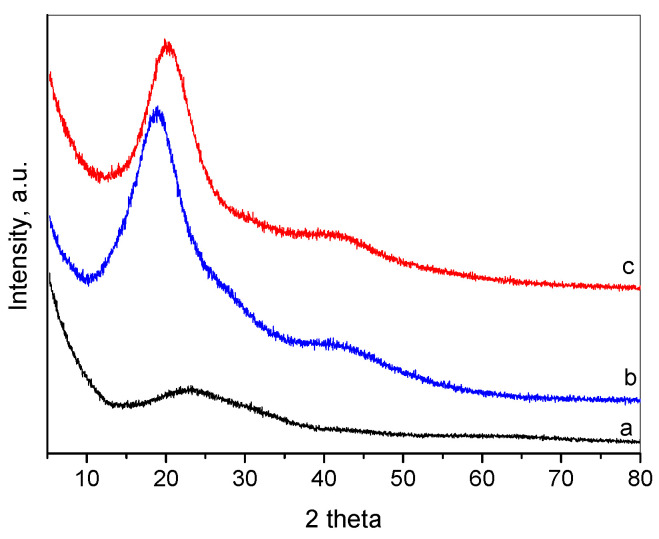
XRD patterns of empty chitosan—alginate nanoparticles (**a**), oregano oil (**b**), and oregano oil-loaded chitosan—alginate nanoparticles (**c**).

**Figure 2 molecules-26-07017-f002:**
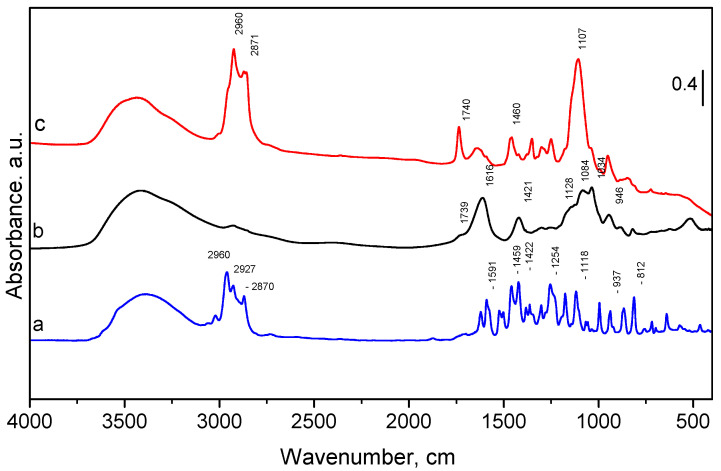
FTIR spectra of pure oregano oil (**a**), empty (**b**) and OrO—loaded chitosan—alginate nanoparticles (**c**).

**Figure 3 molecules-26-07017-f003:**
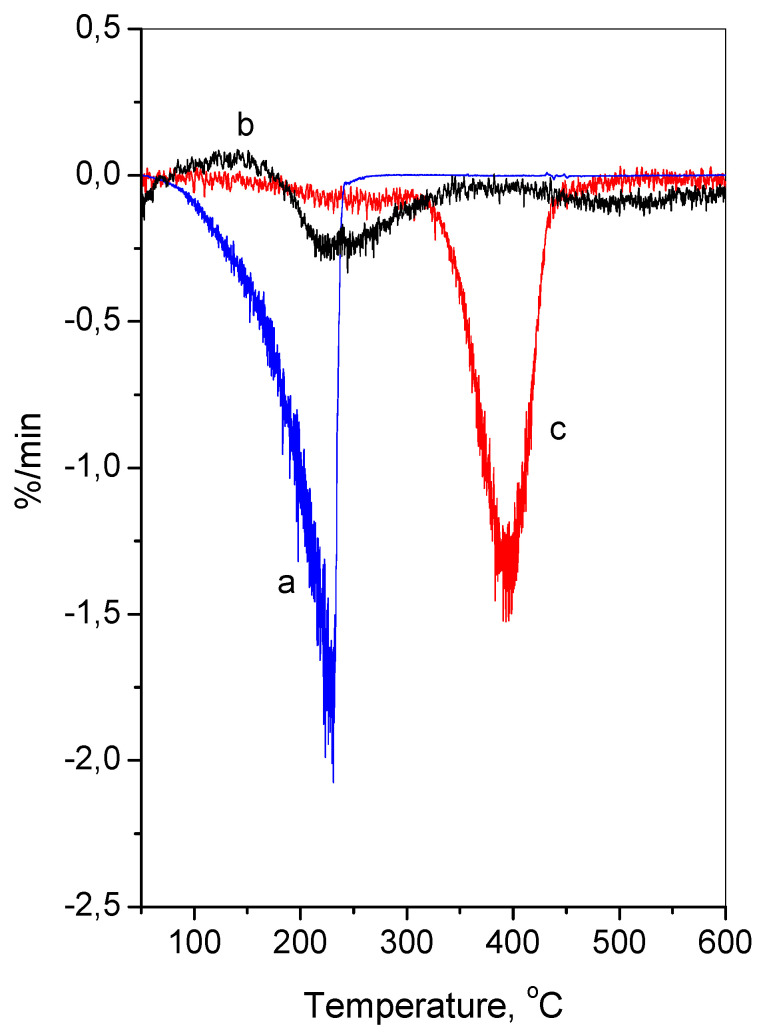
DTG curves of oregano oil (**a**), chitosan–alginate (**b**), and oregano oil–loaded chitosan–alginate (**c**).

**Figure 4 molecules-26-07017-f004:**
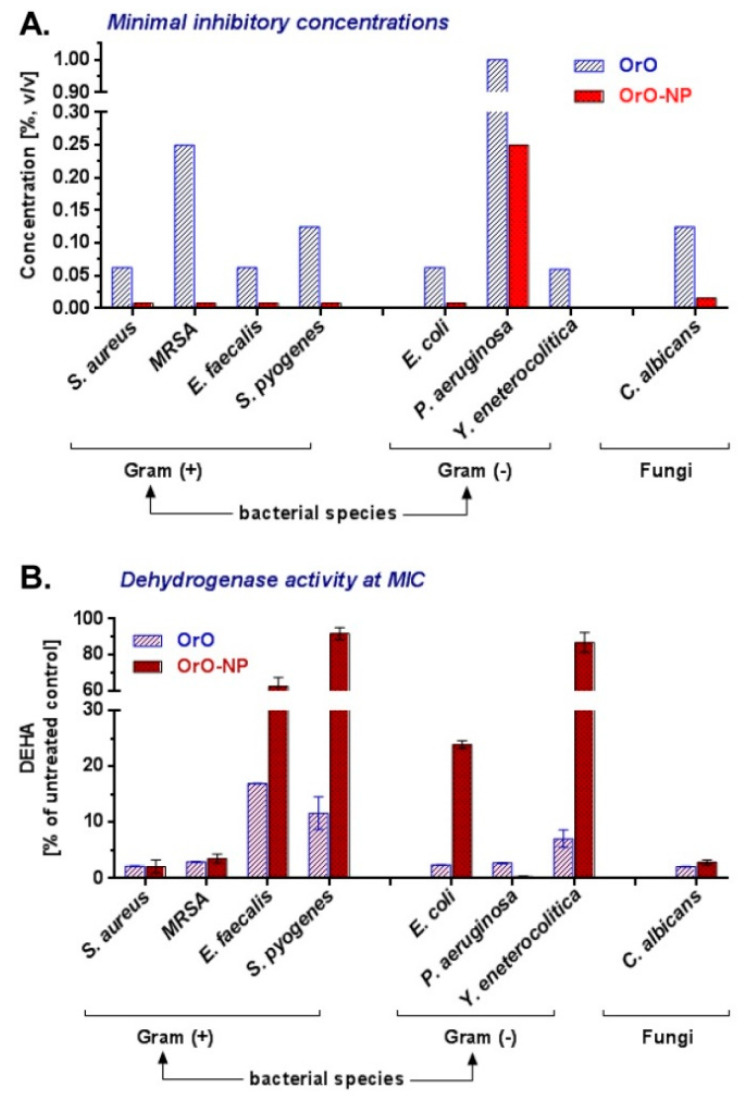
Minimal inhibitory concentrations and dehydrogenase activity at MIC of the oregano oil—comparison between OrO and OrO—NP. Legend: OrO—oregano oil; OrO—NP—OrO—loaded chitosan—alginate nanoparticles.

**Figure 5 molecules-26-07017-f005:**
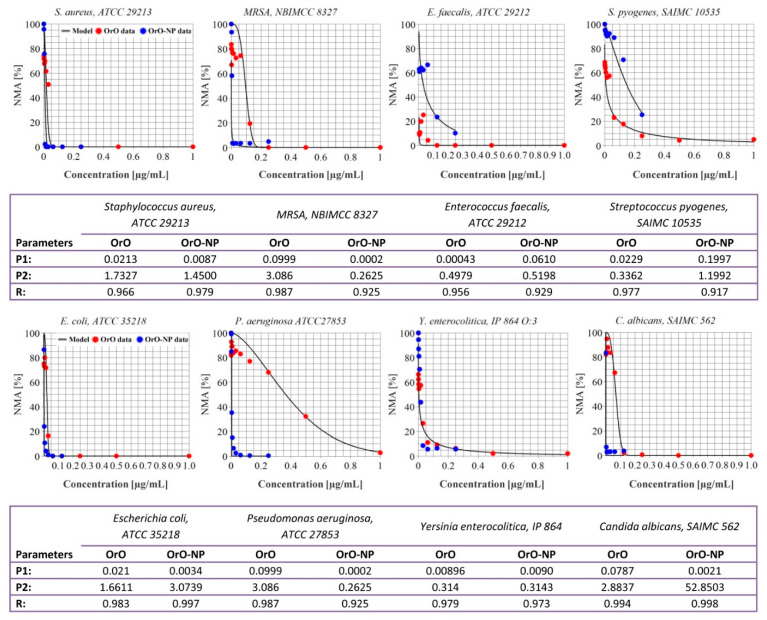
Metabolic activity of the bacterial strains treated with oregano oil—comparison between pure oil and the nanoformulation based on chitosan and alginate. Legend: OrO = *Origanum vulgare* oil, OrO-NP = *Origanum vulgare* oil encapsulated in a chitosan nanodelivery system, P1 = coefficient of inhibition in the Lambert–Pearson model; P2 = hill slope (by realization of the model); R = correlation coefficient showing the descriptive power of the model for the specific experimental data; NMA = normalized metabolic activity.

**Figure 6 molecules-26-07017-f006:**
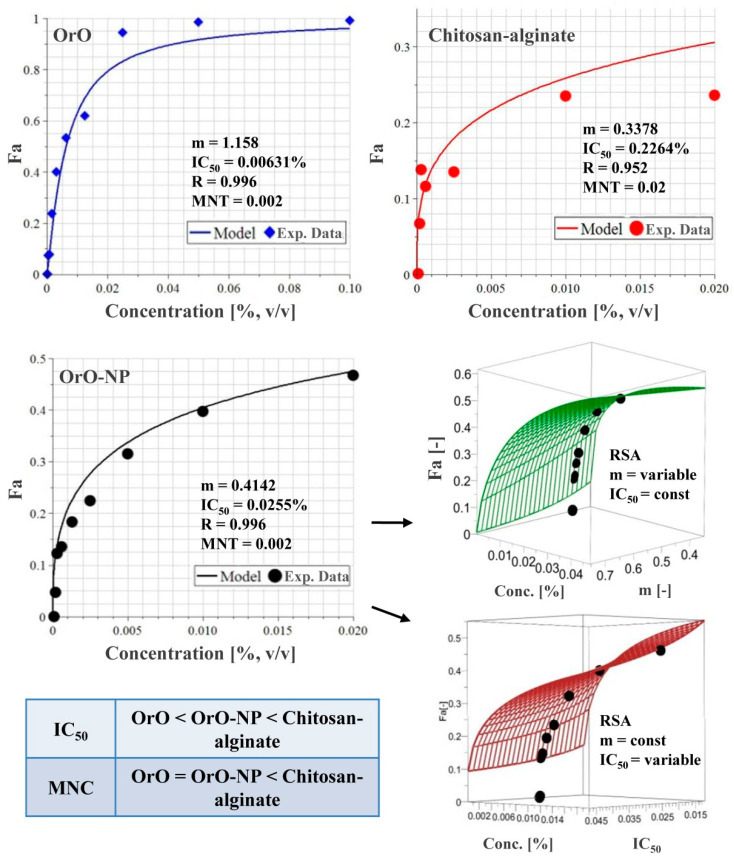
Median inhibitory concentrations, maximum nontoxic concentrations and RSA analysis of oregano oil, chitosan–alginate, and the nanodelivery system of chitosan–alginate loaded with oregano oil. Legend: OrO—*Origanum vulgare* oil, OrO-NP—*Origanum vulgare* oil encapsulated in a chitosan-alginate nanodelivery system.

**Figure 7 molecules-26-07017-f007:**
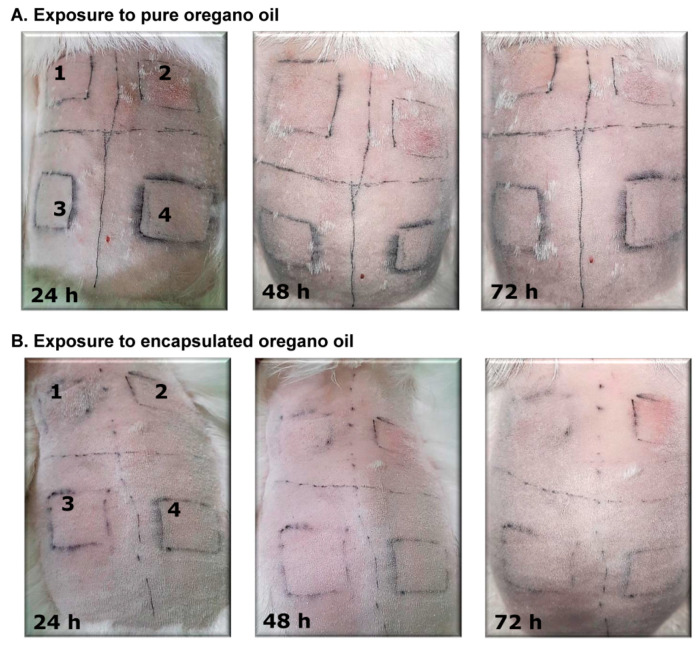
Skin irritation test for pure and encapsulated oregano oil. Legend: 1—administered concentration: 0.1% for pure oregano oil and 0.01% for encapsulated oregano oil; 2—positive control (10% sodium dodecyl sulphate); 3—negative control (sunflower oil); 4—administered concentration: 1% pure oregano oil, 0.1% encapsulated oregano oil.

**Table 1 molecules-26-07017-t001:** GC–MS data of the oregano oil.

Compounds	Rt (min)	Concentration in the Oil (%)
trans-β-ocimene/α-pinene	6.21	1.05
γ-terpinene	8.94	4.05
o-cymene/m-cymene	9.18	39.44
terpinolene	10.28	20.82
bergamol	11.58	n.d.
isothymol methyl ether/carvacrol methyl ether	15.54	n.d.
Thymol	16.88	3.23
Carvacrol	17.08	29.80
aromadendrene	20.10	1.08

**Table 2 molecules-26-07017-t002:** Antimicrobial activity of oregano oil: comparison between pure oil and a nanoformulation based on chitosan and sodium alginate.

Test Microorganisms/Probes	OrO			OrO-NP			Reference Control	
	*MIC* *(%)*	*DEHA* *(%) ± SD*	*MBC* *(%)*	*MIC* *(%)*	*DEHA* *(%) ± SD*	*MBC (%)*	*AB/CT*	*MIC* *(mg/L)*
** *S. aureus* ** **(ATCC 29213)**	0.0625	2.20 ± 0.003	0.25	0.0078	2.12 ± 1.14	0.0625	gentamicin	0.25
**MRSA** **(NBIMCC 8327)**	0.25	2.92 ± 0.03	0.5	0.0078	3.48 ± 0.81	0.125	gentamicin	0.125
** *E. faecalis* ** **(ATCC 29212)**	0.0625	16.94 ± 0.08	0.5	0.0078	62.61 ± 4.81	>0.25 *	penicillin gentamicin	2.5 8
** *S. pyogenes* ** **(SAIMC 10535)**	0.125	11.60 ± 2.90	0.125	0.0078	91.84 ± 3.32	>0.25 *	penicillin	0.08
** *E. coli* ** **(ATCC 35218)**	0.0625	2.40 ± 0.01	0.125	0.0078	23.88 ± 0.70	0.125	gentamicin	2
** *P. aeruginosa* ** **(ATCC 27853)**	1	2.72 ± 0.02	>1	0.25	0.34 ± 0.06	>0.25 *	gentamicin	0.5
** *Y. enterocolitica* ** **(SAIMC 864 O:3)**	0.06	7.05 ± 1.60	0.06	0.002	87.00 ± 5.50	0.25	tetracycline	3
** *C. albicans* ** **(SAIMC 562)**	0.125	2.09 ± 0.06	0.25	0.0156	2.83 ± 0.42	0.25	amphotericin b	0.125

**Legend**: OrO—oregano oil; OrO-NP—oregano oil included in a chitosan-alginate nanodelivery system; MIC—minimal inhibitory concentration; DEHA—dehydrogenase activity; MBC—minimal bactericidal concentration; AB/CT—antibiotic/chemotherapeutic; * the highest tested concentration.

## Data Availability

Not applicable.

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
