# Peer review of "Improvement of the Antimicrobial Activity of Oregano Oil by Encapsulation in Chitosan—Alginate Nanoparticles"

_molecules, 2021, doi:10.3390/molecules26227017_

Round 1

Reviewer 1 Report

The authors present the use of a novel emulsion of oregano oil stabilized by alginate and chitosan. The paper is well written, with a very good flow. I would only advice introducing a paragraph related to the stability of the emulsion (I guess after gelation they are very stable), nevertheless a comment with references is desirable.

Specific comments:

Add value to the discussion with the reference https://doi.org/10.1016/j.ijbiomac.2021.03.084

Line 92 where applicable but its application

Line 111 Add the objective of the paper to introduce the results section. In this work….. or Herein …

Line 161 (Smitha 2005)

Figure 4 indicate which are gram positive and gram negative (maybe separate them in two groups). Quality of Figure should be increased. I would advice something more colorful with a little more effort that supports the high quality of the research

Figure 5 and its discussion should be in the results part. I only find reference of Figure 5 in experimental (Skin irritation test)

Line 275 The authors indicate Nanoparticle diameter, polydispersity in the experimental, I don’t find polydispersity in the results

Line 297 LP meaning

Line 355 Reference for Ph. Eur. 9th

Line 471 Is metabolic activity stated in the results section?

Line 521 P1 meaning

In the conclusions I would advice to indicate in which kind of foods or systems there is a prospect that this formulation could have high antimicrobial activity and protection

Best regards

Author Response

Reviewer 1

The authors present the use of a novel emulsion of oregano oil stabilized by alginate and chitosan. The paper is well written, with a very good flow. I would only advice introducing a paragraph related to the stability of the emulsion (I guess after gelation they are very stable), nevertheless a comment with references is desirable.

Answer: Paragraph related to the stability of the dispersion was added in “Discussion”.

“Thus, the small size and the magnitude of the negative zeta-potential provide good colloidal stability of the dispersion. This is in agreement with reports that values of zeta-potential ±30 mV indicate high stability due to the electrostatic repulsion (Sapsford KE, Tyner KM, Dair BJ, Deschamps JR, Medintz IL. Analyzing nanomaterial bioconjugates: a review of current and emerging purification and characterization techniques. Anal Chem. 2011;83:4453–4488).”

Specific comments:

Comment: Add value to the discussion with the reference https://doi.org/10.1016/j.ijbiomac.2021.03.084

            Answer: the reference is very valuable and it was included in the discussion of the manuscript

Comment: Line 92 where applicable but its application

            Answer: the phrase “where applicable“ was deleted

Comment: Line 111 Add the objective of the paper to introduce the results section. In this work….. or Herein …

            Answer: The following text is added in the revised manuscript: “The aim of our study is to develop a nanosized system that will either enable formulation of oregano oil in aqueous dispersion or potentiate its antimicrobial effect. Chitosan and sodium alginate are selected as vehicles of the nanosytem because of their biocompatibility and antibacterial properties of chitosan itself. Further, part of the studies is focused on evaluation of in vitro cytotoxicity of the developed nanodelivery system in normal human keratinocytes and in vivo skin irritation test on rabbits.”

Comment: Line 161 (Smitha 2005)

            Answer: The reference was numbered.

Comment: Figure 4 indicate which are gram positive and gram negative (maybe separate them in two groups). Quality of Figure should be increased. I would advice something more colorful with a little more effort that supports the high quality of the research

            Answer: Figure 4 was corrected according to the recommendations of the reviewer, namely – the quality of the figure was increaced and all figures are submitted along with the manuscript in jpeg format and 1200 dpi

Comment: Figure 5 and its discussion should be in the results part. I only find reference of Figure 5 in experimental (Skin irritation test)

            Answer: A description of the data on Figure 5 was included in the Results section.

Comment: Line 275 The authors indicate Nanoparticle diameter, polydispersity in the experimental, I don’t find polydispersity in the results

            Answer: The polydispersity index was included in the revised manuscript.

Line 274-276: The oil loaded nanoparticles were characterized with a mean diameter approximately 320 nm, polydispersity index of 0.631and negative zeta-potential (-25 mV).

Comment: Line 297 LP meaning

            Answer: The full name of the model was written.

Comment: Line 355 Reference for Ph. Eur. 9th

            Answer: the reference was included in the bibliography.

Comment: Line 471 Is metabolic activity stated in the results section?

            Answer: Yes, the description of the results was added to the Results section.

Comment: Line 521 P1 meaning

            Answer: P1 is the inhibitory constant in the Lambert-Pearson model as described in section Material and methods and section Results.

Comment: In the conclusions I would advice to indicate in which kind of foods or systems there is a prospect that this formulation could have high antimicrobial activity and protection

            Answer: The conclusion was corrected according to the recommendation of the reviewer and replaced with the following text:

“Chitosan-alginate nanoparticles are considered an appropriate nanosized system for oregano oil. The studies revealed an improved antibacterial activity of encapsulated oregano oil. Interestingly, the encapsulated oregano oil possessed significantly higher antimetabolic activity than the pure oil in 6 of the tested 8 strains. Based on these results the oregano oil-loaded nanosystem is perspective for further development as food additive with antimicrobial activity. In particular, the proposed nanodelivery system should be further screened for antibacterial effect in vacuum packed foods like meat, cheese, etc. accounting for their compatibility to the oregano oil taste and other organoleptic characteristics.

On the other hand, in vitro cytotoxicity on human keratinocytes and in vivo skin irritation test on rabbits demonstrated a safety profile of the nanoparticle formulation. Thus, the developed oregano oil-loaded chitosan-alginate nanoparticles could be considered an appropriate topical delivery system for treatment of microbial infections of skin and other soft tissues.”

Reviewer 2 Report

In this manuscript, the authors reported the incorporation of oregano oil in aqueous dispersion of chitosan-alginate nanoparticles by emulsification and consequent electrostatic gelation of both polysaccharides, and how this will affect its antimicrobial activity. Overall, this is an excellent and well-written article and can arouse wide attention in the field of antimicrobial activity. Therefore, I highly recommend this manuscript to be published. There are only minor comments:

  1. The authors may need to briefly address the difference(s) between the current manuscript and other similar published articles in the Introduction section. Specifically, in addition to polysaccharides being eco-friendly alternatives, why did the authors choose chitosan-alginate nanoparticles over using other polysaccharides? In addition, the authors should add a last paragraph in the Introduction section regarding the main goal of their study.

  1. With regard of the characterization of the chitosan-alginate nanoparticles, the size/morphology of the NPs should be investigated by SEM or TEM if possible. However, if this is not possible, the authors should discuss this point of view: The use of DLS to determine the nanoparticles size has an advantage over non-optical Transmission Electron Microscopy (TEM), which has a much higher resolution but is more expensive and requires special sample preparation. But what about the accuracy of each technique? Since DLS is a technique highly affected by several parameters, such as the impurities in the solution, the concentration….

  1. What was the molecular weight of the starting chitosan and alginate materials used?

  1. In the conclusions, the author should also provide an outlook of the challenges and potential future directions.

Author Response

Reviewer 2

In this manuscript, the authors reported the incorporation of oregano oil in aqueous dispersion of chitosan-alginate nanoparticles by emulsification and consequent electrostatic gelation of both polysaccharides, and how this will affect its antimicrobial activity. Overall, this is an excellent and well-written article and can arouse wide attention in the field of antimicrobial activity. Therefore, I highly recommend this manuscript to be published. There are only minor comments:

1. The authors may need to briefly address the difference(s) between the current manuscript and other similar published articles in the Introduction section. Specifically, in addition to polysaccharides being eco-friendly alternatives, why did the authors choose chitosan-alginate nanoparticles over using other polysaccharides? In addition, the authors should add a last paragraph in the Introduction section regarding the main goal of their study.

Answer: The following text is added in the revised manuscript: “The aim of our study is to develop a nanosized system that will either enable formulation of oregano oil in aqueous dispersion or potentiate its antimicrobial effect. Chitosan and sodium alginate are selected as vehicles of the nanosytem because of their biocompatibility and antibacterial properties of chitosan itself. Further, part of the studies is focused on evaluation of in vitro cytotoxicity of the developed nandelivery system in normal human keratinocytes and in vivo skin irritation test on rabbits.”

2. With regard of the characterization of the chitosan-alginate nanoparticles, the size/morphology of the NPs should be investigated by SEM or TEM if possible. However, if this is not possible, the authors should discuss this point of view: The use of DLS to determine the nanoparticles size has an advantage over non-optical Transmission Electron Microscopy (TEM), which has a much higher resolution but is more expensive and requires special sample preparation. But what about the accuracy of each technique? Since DLS is a technique highly affected by several parameters, such as the impurities in the solution, the concentration….

Answer: We agree with the reviewer, in particular both DLS and SEM (TEM) could be informative about the samples. Microscopic examinations are really more expensive, whereas DLS give also information about polydispersity of the samples. In the present study we included only DLS data taking into account our previous microscopic observations of similar nanosized system (Yoncheva, K., Merino, M., Shenol, A., Daskalov, N. T., Petkov, P. S., Vayssilov, G. N., & Garrido, M. J. (2019) Optimization and in-vitro/in-vivo evaluation of doxorubicin-loaded chitosan-alginate nanoparticles using a melanoma mouse model. Int J Pharm, 556, pp. 1-8.).

3. What was the molecular weight of the starting chitosan and alginate materials used?

Answer: We agree that the molecular weight of both biopolymers is important. We included this information in the section “Materials and methods”.

Line 370-371: Sodium alginate (very low viscosity) was supplied by Alfa Aesar GmBH & CoKG (Karlsruhe, Germany). Chitosan (Mv 110 000 - 150 000) was provided by Sigma Aldrich.

4. In the conclusions, the author should also provide an outlook of the challenges and potential future directions.

Answer: According to the suggestion of the reviewer, we provided an outlook of the challenges and potential future directions.

The conclusion was replaced with the following text:

“Chitosan-alginate nanoparticles are considered an appropriate nanosized system for oregano oil. The studies revealed an improved antibacterial activity of encapsulated oregano oil. Interestingly, the encapsulated oregano oil possessed significantly higher antimetabolic activity than the pure oil in 6 of the tested 8 strains. Based on these results the oregano oil-loaded nanosystem is perspective for further development as food additive with antimicrobial activity. In particular, the proposed nanodelivery system should be further screened for antibacterial effect in vacuum packed foods like meat, cheese, etc. accounting for their compatibility to the oregano oil taste and other organoleptic characteristics.

On the other hand, in vitro cytotoxicity on human keratinocytes and in vivo skin irritation test on rabbits demonstrated a safety profile of the nanoparticle formulation. Thus, the developed oregano oil-loaded chitosan-alginate nanoparticles could be considered an appropriate topical delivery system for treatment of microbial infections of skin and other soft tissues.”